

# Development Processes of the East Asian Cyclones over the Korean Peninsula

Joonsuk M. Kang, Seok-Woo Son

School of Earth and Environmental Sciences, Seoul National University, Seoul, 08826, Republic of Korea

*Correspondence to*: Seok-Woo Son (seokwooson@snu.ac.kr)

**Abstract.** The development processes of the extratropical cyclones passing the Korean Peninsula during the period of 1979–2017 are quantitatively evaluated in the potential vorticity (PV) perspective. A feature tracking algorithm is applied to the ERA-Interim reanalysis data to objectively identify the distinct northern- and southern-track (NT and ST) cyclones affecting the region in the cold season. The dynamic and thermodynamic contributions to the development of these two categories of

cyclones are then comparatively assessed in terms of the relative vorticity tendency resulting from the PV tendency inversion. It is quantified through inversion that the NT cyclones develop 87.9% dynamically and 6.2% thermodynamically. In contrast, the respective contributions to the ST cyclones are 71.8% and 43.5% for the ST cyclones, with negative effects from non-explicit processes. In both NT and ST cyclones, the zonal PV advection in the upper troposphere is the most influential for the dynamic development, while nonlinear advection being more important in the former. The larger thermodynamic

contribution of the latter is attributed to more latent heating being involved in the development, which produces more lower-level PV and reduces damping from vertical PV advection. These results indicate that East Asian cyclones passing the Korean Peninsula have different development processes depending on their tracks.

## 1 Introduction

The extratropical cyclones (ETCs) are, to a great extent, responsible for midlatitude surface weather and its extremes (Pfahl

and Wernli, 2012; Bentley et al., 2019). The region of frequent ETC occurrence, referred to as stormtracks, locates over the Pacific and Atlantic Oceans in the Northern Hemisphere (Chang et al., 2002; Hoskins and Hodges, 2002). This ETC development over the oceans is facilitated by the surface fluxes and strong baroclinicity imposed by the sharp sea surface temperature gradient, particularly in winter.

Though less prominent in both features, ETCs are also frequently observed over the continents of East Asia and North

America (Plante et al., 2015; Lee et al., 2019). The formation of these continental ETCs is often related to upstream orography (Chung et al., 1976; Whittaker and Horn, 1984). Unlike their oceanic counterparts, continental ETCs pass through highly-populated regions while they travel eastward. In this sense, understanding the mechanisms responsible for the intensification and propagation of these cyclones is crucial for disaster management.

In East Asia, cyclogenesis is remarked over Mongolia and East China (Chen et al., 1991; Adachi and Kimura, 2007; see Fig.

3 in Lee et al., 2019). The ETCs from the two regions have somewhat similar temporal characteristics with a maximum



frequency and intensity in spring (March–May) during the year (Lee et al., 2019). However, their development processes, especially as they travel past the Korean Peninsula, are different (Kang et al., 2020; hereafter K20). In K20, the synoptic structures of the two ETC groups throughout the development stages are investigated by performing a composite analysis of objectively-tracked cyclones. It is found that the vertical distribution of geopotential anomaly significantly differs between
the two groups. While the northern-track (NT) cyclones from Mongolia have negative geopotential anomalies maximized around the tropopause, the southern-track (ST) cyclones from East China have maximum negative anomalies near the surface (see Figs. 6 and 10 in K20). This result implies that the NT cyclones are mostly driven by the upper-level processes, whereas the ST cyclones are more strongly influenced by the lower-level processes.

The distinct vertical structure can be explained in the potential vorticity (PV) perspective (Hoskins et al., 1985). The positive
upper-tropospheric PV anomaly is strong in NT cyclones (see Fig. 7 in K20), evidencing the influence of baroclinic instability. However, the same anomaly is weak or nearly absent for ST cyclones. Instead, the lower-tropospheric PV anomaly is strong (Fig. 11 in K20), possibly due to latent heating (LH).

The qualitative PV analysis in K20 provides a hint on the development mechanism of the two groups of ETCs in the region. Nevertheless, further analyses are required to quantify the contributions from, for instance, the upper-level trough and LH.
Such quantifications are particularly important for the strong NT and ST cyclones. During the development, the NT cyclones may also have a large lower-tropospheric PV anomaly as ST cyclones. The development processes, therefore, become less discernible through simple composite maps. Additionally, less attention is paid to the role of background flow in K20, which is expected to have a significant and primary effect on ETC development (Eady, 1949; Tierney et al., 2018; Kang and Son, 2020).

To this end, the qualitative analysis in K20 is quantitatively extended in this study by considering cold season ETCs from October to May. The contributions of upper- and lower-level PV processes, LH, and surface conditions to the ETC development and their differences between NT and ST cyclones are evaluated using the PV tendency inversion (Kang and Son, 2020; hereafter KS20). This method, analogous to the PV inversion (Davis and Emanuel, 1991), calculates geopotential tendency from local PV tendency. The linear partitioning of this local PV tendency as in piecewise PV inversion (Davis,
1992) is made with the PV tendency equation (e.g., Tamarin and Kaspi, 2016). With this method, particular attention is paid to the effect of mean and anomalous flows and their diverse interactions.

The rest of the paper is organized as follows. Section 2 describes the data and method, including the PV tendency inversion. The synoptic structure of NT and ST cyclones are investigated in Sect. 3. The results from the inversion calculations are scrutinized in Sect. 4. Finally, summary and discussion are presented in Sect. 5.

**2 Data and Methods**



## 2.1 Data

The six-hourly ERA-Interim dataset (Dee et al., 2011), interpolated onto 1.5°×1.5° latitude-longitude grid and 37 pressure
levels, are utilized. The horizontal winds, relative vorticity, temperature, geopotential, specific humidity, and pressure
velocity data during 39 years (1979–2017) are used as in K20. The PV is calculated from horizontal winds and temperature
by approximating partial differentials with second-order finite differencing.
.

## 2.2 ETC tracking and selection

For the objective identification and tracking of ETCs, the Hodges (1995,1999) algorithm is employed. In this method, 850-
hPa relative vorticity data from wavenumber 5 to 42 is utilized. The spatial filtering is made to focus on synoptic-scale
circulation, and the 850-hPa pressure level is chosen to minimize errors from extrapolation owing to the elevated topography
over East Asia (Lee et al., 2019). From this vorticity filed, the method identifies the vorticity maxima exceeding 1 Cyclonic
Vorticity Unit (CVU; 1 CVU = $10^{-5}$ s$^{-1}$). The detected maxima are grouped and regarded as ETC tracks if their migrations
during consecutive six-hourly time steps satisfy the criteria imposed by the algorithm, based on their propagation speed and
direction. Among the ETC tracks, short-lived and stationary cyclones are excluded by considering tracks that last longer than
48 hours and travels farther than 1000 km. Only the ETCs generated poleward of 25°N are considered, to discard tropical
cyclones.

From the tracking algorithm, the ETCs passing the Korean Peninsula (120–135°E, 33–48°N; yellow box in Fig. 1) during the
period of 1979–2017 is selected. Their track density in the cold season (October–May) is illustrated in Fig. 1a. More than 25
ETCs impact the region in each along the two distinct ETC tracks. As noted in the previous section, they are mostly initiated
over Mongolia and East China (black contours in Fig. 1a). The ETCs originating from these two regions are objectively
classified, using the fuzzy *c*-means clustering (Bezdek et al., 1984), into northern-track (NT) and southern-track (ST)
cyclones, respectively.

With the focus on the development processes, the cold season ETCs with the top 10% *maximum intensification rate in the
target domain* are considered for the quantitative analyses. Here, the intensification rate is calculated as the 12-hour
difference of relative vorticity, and the time of its maximum in the target domain is defined as $t_{max}$ (filled squares in Fig. 1b).
These ETCs account for 88 NT and 106 ST cyclones. The average tracks of these NT and ST cyclones are depicted in Fig.
1b, where both tracks exhibit northeastward propagation after $t_{max}$. These top 10% ETCs are simply referred to as NT and ST
cyclones hereafter. Note that these ETCs fall into the categories of rapidly intensifying cyclones in K20..






**2.3 PV tendency inversion**

To quantitatively assess the ETC development processes, the PV tendency inversion method (KS20) is utilized in this study. This method allows a prognostic quantification of ETC development as in Zwack-Okossi equation (Zwack and Okossi, 1986) or pressure tendency equation (Fink et al. 2012), but in PV perspective. In this method, the processes responsible for ETC

development are described through the PV tendency equation, which is expressed as

$$\frac{\partial q}{\partial t} = -\mathbf{v} \cdot \nabla q - \omega \frac{\partial q}{\partial p} + Q_{LH} + F_{RES}. \tag{1}$$

In Eq. (1), $q$ is PV, $\mathbf{v} = (u, v, 0)$ is the horizontal wind vector, $\nabla = \left(\frac{\partial}{\partial x}, \frac{\partial}{\partial y}, 0\right)$ is the horizontal gradient operator, and $\omega$ is the pressure velocity. The first term on the rhs of Eq. (1), representing the horizontal PV advection, describes the effects of the propagation and interaction of the upper-level trough and lower-level cyclonic circulation. The second term, representing

the vertical PV advection, is physically related to the vertical change of adiabatic cooling which generally weakens ETC development. The third and fourth terms respectively stand for local PV changes from latent heating ($Q_{LH}$) and other non-conservative processes ($F_{RES}$) such as friction and cloud radiation. The formulation of $Q_{LH}$ follows that in Tamarin and Kaspi (2016), and the latent heating is calculated as in Emanuel et al. (1987).

To investigate ETC development in detail, the advection terms are decomposed as follows.

$$-\mathbf{v} \cdot \nabla q - \omega \frac{\partial q}{\partial p} = -\overline{\mathbf{v}} \cdot \nabla q - \mathbf{v}' \cdot \nabla \overline{q} - \mathbf{v}' \cdot \nabla q' - \omega \frac{\partial \overline{q}}{\partial p} - \omega \frac{\partial q'}{\partial p} \tag{2}$$

Here, the overbar denotes the mean, which is the monthly climatology during the analysis period, and the prime stands for the deviation from that value. Unlike in KS20, the mean value is not zonally averaged to account for the zonally varying background flow on ETC development (e.g., Tamarin and Kaspi 2017). Note that the advection by the mean wind is collectively considered in Eq. (2) because $-\overline{\mathbf{v}} \cdot \nabla q$ is dominated by $-\overline{\mathbf{v}} \cdot \nabla q'$ with a negligible contribution from $\overline{\mathbf{v}} \cdot \nabla \overline{q}$.

To calculate the circulation changes that the rhs terms in Eq. (1) make, the lhs of Eq. (1) is approximated to a linear function of geopotential tendency ($\chi$) as

$$\frac{\partial q}{\partial t} \approx \mathrm{L}(\chi). \tag{3}$$

Here, $\mathrm{L} \equiv -g \frac{\partial \overline{\theta}}{\partial p}\left(\frac{1}{f_0}\nabla^2 + \frac{f}{\sigma}\frac{\partial^2}{\partial p^2}\right)$, where $g$ is the gravitational acceleration, $\theta$ is the potential temperature, $f$ is the Coriolis parameter with $f_0$ at ETC center, $\sigma = -\frac{R_d \overline{T}}{p\overline{\theta}}\frac{\partial \overline{\theta}}{\partial p}$, $R_d$ is the gas constant for dry air, and $T$ is temperature. By inverting the

operator L, Eq. (3) can be expressed as

$$\chi \approx \mathrm{L}^{-1}\left(\frac{\partial q}{\partial t}\right), \tag{4}$$

showing that geopotential tendency is achievable from a local PV tendency. It is shown in KS20 that approximations made in Eqs. (3) and (4) are fairly valid, and the circulation obtained from inversion well represents the observed state (see Figs. 4 and 8 in KS20). Furthermore, the linearity of the operator suggests that the rhs of Eq. (1) serves as linear partitions of $\frac{\partial q}{\partial t}$,

leading to the following equation.





$$L^{-1}\left(\frac{\partial q}{\partial t}\right) = L^{-1}(-\mathbf{v} \cdot \nabla q) + L^{-1}\left(-\omega \frac{\partial q}{\partial p}\right) + L^{-1}(Q_{LH}) + L^{-1}(F_{RES}) + \chi_{BC} \tag{5}$$

The rhs terms indicate the geopotential tendency induced by respective ETC development processes. The last term, $\chi_{BC}$, represents the geopotential tendency response to the boundary condition used.

For the boundary condition, homogeneous Dirichlet boundary condition ($\chi = 0$) is applied at the lateral boundaries. At the

top and bottom boundaries, the Neumann boundary conditions are used as below.

$$\frac{\partial \chi}{\partial p} = -\frac{R_d}{p} \frac{\partial T}{\partial t} \tag{6}$$

The rhs of Eq. (6) is calculated at a level interior of the top and bottom surfaces (175 and 875 hPa) and used only in the calculation of $L^{-1}\left(\frac{\partial q}{\partial t}\right)$ and $\chi_{BC}$ and is set to zero otherwise. All other details of the inversion are in compliance with those in KS20. In this way, inversions in Eq. (5) are carried out for all ETCs that are selected in the following section. Additionally,

$L^{-1}(-\mathbf{v} \cdot \nabla q)$ and $L^{-1}\left(-\omega \frac{\partial q}{\partial p}\right)$ are examined in detail according to Eq. (2).

## 3. Background Flow and PV Structure of ETCs

### 3.1 Background flow

Prior to investigating individual ETCs, the background flow which affects the ETC development is discussed here. The cold

season climatology of PV and wind at 250 hPa is shown in Fig. 2a. A stationary trough is located over the Okhotsk Sea. The associated PV gradient is large equatorward to the trough. The mean wind is mostly normal to the PV gradient, and a strong jet locates south of the Korean Peninsula. Though not shown, the lower-level climatological PV distribution resembles that in the upper troposphere, but with a weaker meridional gradient.

The lower-level mean wind over and upstream of the Korean Peninsula is northwesterly (Fig. 2b), likely responsible for the

southeastward propagation of the NT cyclones. However, it would hinder the northeastward propagation of ST cyclones at the same time. This indicates that the ETC propagation does not solely depend on the steering of the mean wind. The climatological water vapor, integrated from 1000 to 300 hPa, is also shown in Fig. 2b. As sharp moisture gradient locates over 30°N, the tropospheric moisture is about twice more abundant in the cyclogenesis region of ST cyclones compared to that of NT cyclones.


### 3.2 PV structure of NT and ST cyclones

Figure 3 illustrates the PV and wind anomalies of NT and ST cyclones at $t_{max}$. As mentioned above, the anomalies are deviations from monthly climatology during the 39 years (1979–2017). The NT cyclones accompany an enhanced upper-level trough, represented by 250-hPa PV anomaly greater than 3 Potential Vorticity Unit (PVU; 1 PVU = $10^{-6}$ K m$^2$ s$^{-1}$ kg$^{-1}$)



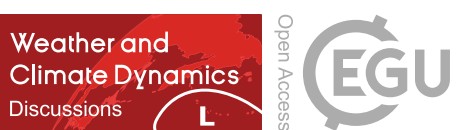

(Fig. 3a). The upper-level trough is weaker for ST cyclones, where a PV anomaly of about 2.5 PVU is found in a narrower region (Fig. 3b). On the contrary, the downstream negative anomaly is stronger in ST cyclones. It is also notable from Figs. 3a and b that the shape of the PV anomalies slightly differs between NT and ST cyclones. While a positive (NE–SW) tilt is found in NT cyclones, a neutral-to-negative (NW–SE) tilt is observed in ST cyclones. This contrast results in different anomalous wind patterns in the upper troposphere (compare Figs. 3a and b).

In the lower troposphere, PV anomalies of ST cyclones are twice as stronger as those of NT cyclones (compare Figs. 3c and d). Intense lower-level PV anomalies for ST cyclones could be attributed to large LH in the mid-troposphere (see Fig. 10 in K20), which is also responsible for the negative anomalies in the upper level (Fig. 3b). Cyclonic winds are found in both cyclones, but southerly on the east of ETC center is stronger in ST cyclones. However, unlike their distinctive PV structures and magnitudes, the intensities of NT and ST cyclones at $t_{max}$ are rather comparable. The former and latter are 4.9 and 5.4

CVU, respectively (not shown).

The results shown in Fig. 3 are largely consistent with those reported in K20, who examined only springtime ETCs. This implies that the development processes of NT and ST cyclones do not have distinct sub-seasonality during the cold season. The overall effect of the PV structures on ETC development can be evaluated by the terms on the rhs of Eq. (1) and their inversions as discussed in Sect. 4.

Figures 4a and b present the temporal tendency of the PV anomalies shown in Fig. 3 as a function of longitude and pressure. In both figures, a westward-tilted positive PV tendency structure is evident. At the same time, a negative PV tendency structure with a weaker magnitude is found to the west. This pattern indicates not only the eastward propagation of ETCs, but also the concurrent amplification. Focusing on the positive tendency, it is comparable in the upper troposphere for NT and ST cyclones. However, the ST cyclones exhibit a stronger tendency in the lower troposphere than the NT cyclones

(compare Figs. 4a and b).

The positive PV tendency in the upper troposphere is mostly due to $-\mathbf{v} \cdot \nabla q$ (Figs. 4c and d). The positive tendency from $-\mathbf{v} \cdot \nabla q$ is stronger for NT (5.6 PVU (12h)$^{-1}$) than ST (4.7 PVU (12h)$^{-1}$) cyclones, implying that ETC development would likely be more related to upper-level waves in the former. However, it is partly canceled out by the negative tendency from $-\omega \frac{\partial q}{\partial p}$, which is slightly stronger for NT cyclones (Figs. 4e and f). The lower-tropospheric portion of positive PV tendency is

primarily affected by $Q_{LH}$, the PV production from LH (Figs. 4g and h). It is about 1.5 times stronger in ST cyclones, consistent with larger lower-level PV anomalies shown in Fig. 3c. In both cases, these tendencies are weakened in the lowermost troposphere and reinforced in the mid-troposphere by $-\omega \frac{\partial q}{\partial p}$ (Figs. 4e and f).



## 4. PV Tendency Inversion

### 4.1 Inversion results

The PV tendencies in Fig. 4 produce the respective geopotential tendencies when inverted. To evaluate the ETC development from the inverted field, the relative vorticity tendency ($\zeta_t$) is achieved from geopotential tendency as

$$\zeta_t = \frac{1}{f_0} \nabla^2 \chi.$$

The resulting 850-hPa vorticity tendency is depicted in Fig. 5. From the inversion of $\frac{\partial q}{\partial t}$, a positive tendency is found to the

northeast of the ETC center (Figs. 5a and b). Focusing on this positive tendency, which is the approximate location of the ETC at $t_{max}+6$ h, the maximum tendency is about 9.2 and 10.7 CVU $(12h)^{-1}$ for NT and ST cyclones, respectively.

A leading contributor to this tendency is the horizontal PV advection, $-\mathbf{v} \cdot \nabla q$ (Figs 5c and d), which produces a positive tendency east to the ETC center. Primarily, the horizontal advection induces positive $\zeta_t$ by propagating the ETC in the lower level. However it is not solely affected by propagation. The positive $\zeta_t$ is also a result of an amplifying upper-level trough.

The fact that positive $\zeta_t$ on the east is stronger than negative $\zeta_t$ on the west indicates that the upper-level trough is amplifying compared to the ridge (see Figs. 4c and d).

The $\zeta_t$ from $-\mathbf{v} \cdot \nabla q$ is slightly stronger in ST cyclones than in NT cyclones. This difference is somewhat the opposite of what is shown in Figs. 4c and d, where $-\mathbf{v} \cdot \nabla q$ is larger for NT cyclones in the upper troposphere. However, it is important to note that ST cyclones have greater $-\mathbf{v} \cdot \nabla q$ at the lower troposphere. This result clearly indicates that diagnosing the PV

tendency at a single level as in Figs. 4c and d is insufficient at gauging its effect on ETC development, highlighting the advantage of inversion calculation.

The contribution of LH on the ETC development, quantified through the inversion of $Q_{LH}$, exhibits a significant difference between NT and ST cyclones as expected (Figs. 5g and h). The positive tendency of 2.6 CVU $(12h)^{-1}$ is induced to the east of NT cyclones' center, whereas that of 4.5 CVU $(12h)^{-1}$ is found in ST cyclones. The difference is also found from the

damping effect by $-\omega \frac{\partial q}{\partial p}$ (Figs. 5e and f). The negative tendency is brought in a broader region for NT cyclones, but occurs in a narrower region with a smaller amplitude for ST cyclones. The surface temperature tendency produces a dipole $\zeta_t$ pattern (Figs. 5i and j), associated with eastward propagating warm and cold sectors (not shown). However, its phase leads the phase of $\zeta_t$ from $\frac{\partial q}{\partial t}$ (Figs. 5a and b), implying that the surface condition may not act favorably for the development of both NT and ST cyclones.

To more quantitatively describe the roles of the relative importance of each process, a 6°×6° box, centered at the maximum $\zeta_t$ from $\frac{\partial q}{\partial t}$ is set for each ETC (see Figs. 5a and b). Then, the $\zeta_t$ inside the box are averaged for all inversion calculations. As discussed in KS20, this area-averaged $\zeta_t$ not only includes the effect of ETC intensification but also that of propagation, allowing for a comprehensive quantification of ETC development. The area-averaged $\zeta_t$, simply referred to as $\zeta_t$, is shown in Fig. 6. Note that since the exact location of the box varies for each ETC, the $\zeta_t$ averaged for all analyzed ETCs is slightly





different from the area-averaged $\zeta_t$ from the composite field. It turns out that the $\zeta_t$ from $\frac{\partial q}{\partial t}$ is 4.9($\pm$0.5) and 5.9($\pm$0.5) CVU

(12h)$^{-1}$ for NT and ST cyclones, respectively (left black). With slight over/underestimations, the method well reproduces $\zeta_t$

from the reanalysis (pink lines) within the 95% confidence intervals. The linearity is also well kept during the inversion

processes (compare leftmost and rightmost black bars), allowing for term-by-term analysis that follows.

As anticipated, $-\mathbf{v} \cdot \nabla q$ makes the most pivotal contribution to the NT cyclone development (red; 4.3 CVU (12h)$^{-1}$),

explaining 87.9% of total development (Fig. 6a). The $Q_{LH}$ is responsible for about one-fourth of the development (cyan; 1.3

CVU (12h)$^{-1}$), but is obstructed to a large extent by $-\omega \frac{\partial q}{\partial p}$ (yellow; -0.9 CVU (12h)$^{-1}$). The effect of phase-shifted surface

temperature tendency is insignificant (brown; -0.2 CVU (12h)$^{-1}$).

The largest contributor to the ST cyclones' development is again $-\mathbf{v} \cdot \nabla q$, which increases $\zeta_t$ by 4.2 CVU (12h)$^{-1}$ (red; Fig.

6b). Second to $-\mathbf{v} \cdot \nabla q$, the $\zeta_t$ from $Q_{LH}$ explains 48.7% of the cyclone development (cyan; 2.9 CVU (12h)$^{-1}$). The damping

effects from $-\omega \frac{\partial q}{\partial p}$ and $-\frac{R_d}{p} \frac{\partial T}{\partial t}$ are -0.3 and -0.5 CVU (12h)$^{-1}$ (yellow and brown), respectively, and the former is

statistically insignificant.

Comparing Figs. 6a and b, each process exhibits a substantial difference in terms of both absolute and relative contributions

to the development of NT and ST cyclones. For instance, the absolute contribution of $-\mathbf{v} \cdot \nabla q$ is comparable in both cases.

However, its relative impact is quite different. While it accounts for 87.9% of NT cyclone development, it explains only 71.8%

of ST cyclone development. The stark difference in LH influence, as indicated in K20 and Figs. 3 and 4 of this study, is also

evident. The $\zeta_t$ from $Q_{LH}$ is 1.6 CVU (12h)$^{-1}$ stronger in ST cyclones (23% larger relative contribution), likely responsible

for the larger total $\zeta_t$ of ST cyclones. It is worth to note that the negative effect of $-\omega \frac{\partial q}{\partial p}$ typically emerges from PV

stratification ($\frac{\partial q}{\partial p} < 0$), which roughly corresponds to the vertical change of static stability ($-g \frac{\partial \theta}{\partial p}$). This process significantly

weakens the development of NT cyclones, but its effect is one-third in ST cyclones.

By referring $Q_{LH}$ and $-\omega \frac{\partial q}{\partial p}$ together as thermodynamical processes and $-\mathbf{v} \cdot \nabla q$ as dynamical process in the free

atmosphere, it is concluded that the NT cyclones are mostly dynamically driven (87.9%). The thermodynamic contribution is

rather minor and smaller than the non-explicit processes. The ST cyclones develop under both dynamical (71.8%) and

thermodynamical (43.5%) influences. A strong thermodynamic contribution, much stronger than that in NT cyclone, is

mostly due to a large amount of environmental moisture at the genesis region (Fig. 2b).

As documented in K20, the structures of NT and ST cyclones not only differ in their rapid intensification phase (t$_{max}$) but

also in the initial stages. In this regard, additional inversions are performed while retroceding to t$_{max}$−36 h in six-hourly

intervals. Note that at t$_{max}$−36 h (−36 h for brevity hereafter), the numbers of analyzed ETCs are only 83% and 49% of those

at t$_{max}$ for NT and ST cyclones, respectively (73 and 52 ETCs), as some cyclones reach t$_{max}$ in less than 36 hours from their

formation.

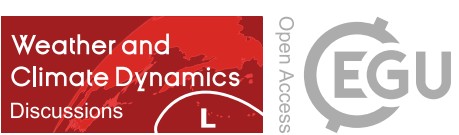

The results of these additional inversions are illustrated in Fig. 7. At initial stages, the NT cyclones feature a larger total $\zeta_t$ than ST cyclones, unlike at $t_{max}$. The dynamical process is still the major contributor to the initial development of NT cyclones (Fig. 7a; ~75%). The contribution of $Q_{LH}$ is only about 17% of the total development at −36 and −30 h and does not exceed above 30% during the 36 hours. On the contrary, the thermodynamic contribution plays a critical role in the development for ST cyclones in the early stages (−36 to −24 h), being comparable to or even larger than the dynamic

contribution. Throughout the 36 hours, $\zeta_t$ from $Q_{LH}$ does not fall below 45% of the total development. This persistent contribution of $Q_{LH}$ further explains how ST cyclones propagate northeastward, as opposed to the southeastward mean flow (Fig. 2b). As $\zeta_t$ from $Q_{LH}$ increases drastically from −18 h, the total $\zeta_t$ also rises sharply to surpass that of NT cyclones.

During the 36 hours, the effect of $-\omega\frac{\partial q}{\partial p}$ on the development of NT cyclones is insignificant, except at near $t_{max}$. The same process positively contributes to the development of ST cyclones in their early stages, although hardly significant. As in $t_{max}$,

the temperature tendency at the surface mainly has a negative influence on the development throughout the 36 hours for both ETCs.

## 4.2 Decomposition of advection terms

While the results shown in Figs. 6 and 7 give quantitative analyses on the ETC development by evaluating the terms on the

rhs of Eq. (1), further insight can be achieved by decomposing the advection terms. As shown in the rhs of Eq. (2), the horizontal advections can be divided into zonal and meridional components with mean and anomaly components and upper- and lower-tropospheric portions across 600 hPa (KS20).

Figure 8 depicts the $\zeta_t$ from the decomposed advection terms. Before comparing NT and ST cyclones, it is worth addressing the general aspects of ETC development in this region. The positive contribution by the horizontal PV advection ($-\mathbf{v}\cdot\nabla q$) is

mostly from the zonal advection ($-u\frac{\partial q}{\partial x}$). The meridional advection ($-v\frac{\partial q}{\partial y}$) acts to damp it (Figs. 8a–d). Generally, the positive and negative effects by $-\mathbf{v}\cdot\nabla q$ is derived more from the upper troposphere, due to stronger PV gradients and winds. In both NT and ST cyclones, a considerable positive contribution is made from $-\overline{u}\frac{\partial q}{\partial x}$ (~5.5 CVU (12h)$^{-1}$). This process is pronounced in the upper level due to the advection of the amplifying trough. Though the PV anomalies in the upper level are stronger for NT cyclones (compare Figs. 3a and b), the $\zeta_t$ from upper-level $-\overline{u}\frac{\partial q}{\partial x}$ is larger for ST cyclones. This is related to

the enhanced downstream ridge and neutral tilt of trough for ST cyclones that experience slightly stronger $\overline{u}$ (Fig. 2; larger to the south) and larger $-\frac{\partial q'}{\partial x}$. The $\zeta_t$ from lower-level $-\overline{u}\frac{\partial q}{\partial x}$ likely represents the propagation of ETCs, and it is larger for ST cyclones, which have larger PV anomaly at $t_{max}$ (Figs. 3c and d).





The upper-tropospheric northerly mean wind ($\overline{v} < 0$) advects PV anomaly southward, as opposed to the direction of cyclone propagation (Fig. 1b). Thus, $-\overline{v}\frac{\partial q}{\partial y}$ in the upper level has a weak negative effect on the ETC development in both cases (Figs.
8c and d). The same process in the lower troposphere does not have significant impact on the ETC development.

While $\zeta_t$ from $-u'\frac{\partial \overline{q}}{\partial x}$ is negligible (Figs. 8a and b), the $-v'\frac{\partial \overline{q}}{\partial y}$ weakens both NT and ST cyclones by introducing low-PV air into ETC center (Figs. 8c and d). This process is more prominent in the upper-troposphere, and its impact on the ETC development is larger for ST cyclones with a stronger southerly wind (compare Figs. 3a and b). It is also evident that the role of upper-level nonlinear interaction terms ($-u'\frac{\partial q'}{\partial x}$ and $-v'\frac{\partial q'}{\partial y}$) are different between NT and ST cyclones. By the positively
tilted trough, the nonlinear processes, especially $-u'\frac{\partial q'}{\partial x}$, foster the development of NT cyclones. The ST cyclones, without a tilted upper-level trough, are not significantly affected by this process. In the lower-level, the nonlinear advections tend to weaken the development of both NT and ST cyclones.

In the vertical (Figs. 8e and f), the climatological PV stratification ($-\omega\frac{\partial \overline{q}}{\partial p}$) damps ETC development, but the altered stratification cancels it ($-\omega\frac{\partial q'}{\partial p}$). The effects from $-\omega\frac{\partial \overline{q}}{\partial p}$ and $-\omega\frac{\partial q'}{\partial p}$ are greater for the ST cyclones due to larger $\omega$ (not
shown). However, because the LH process produces PV anomalies in the lower level, the PV stratification is considerably reduced and large cancellation occurs between $\zeta_t$ from $-\omega\frac{\partial \overline{q}}{\partial p}$ and $-\omega\frac{\partial q'}{\partial p}$.

## 5. Summary and Discussion

In this study, the development processes of East Asian ETCs that pass through and develop over the Korean Peninsula are
quantitatively assessed, extending the qualitative analysis conducted in K20. The PV tendency inversion (KS20) is utilized to evaluate the ETC development caused by the terms in the PV tendency equation. Consistent with previous knowledge, the ETCs from Mongolia (NT cyclones) are mostly dynamically driven. The ETCs originating from East China (ST cyclones) are characterized by strong moist/thermodynamic processes involved along with dynamical processes. Quantitatively, dynamical and thermodynamical processes account for 87.9% and 6.2% of NT cyclone development but 71.8% and 43.5%
of ST cyclone development at $t_{max}$, with other minor positive/negative contributors. This quantitative difference between NT and ST cyclone developments exist from their initial stages and is even clearer at then than at $t_{max}$.

Additional inversions with decomposed PV advections highlight the role of upper-level mean flow on the dynamic development of the ETCs in this region. The mean zonal wind advects the upper-tropospheric PV anomaly over the ETC center and intensifies it (KS20). The results also suggest that the upper-level processes could further foster development
through nonlinear advections, when the upper-level trough has a positive tilt as in NT cyclones. In fact, this tilt is favorable for the barotropic conversion of eddy kinetic energy on the poleward flank of the jet.

The difference in the thermodynamic contribution to the development between NT and ST cyclones can be primarily related to the difference in the LH involved in the development, due to differing environmental moisture over respective tracks. However, the difference is further enhanced by the altered vertical PV distribution from LH. As such, the damping effect from vertical PV advection is minor for ST cyclones. Moreover, the vertical PV advection favors the development in the initial stages of the ST cyclones. This exemplifies how a strong LH can maintain a cyclone only with small dynamic contributions (e.g., diabatic Rossby waves; Boettcher and Wernli 2013).

The analyses in this study allude that the PV tendency inversion serves as an appropriate tool in quantifying the difference in the development of NT and ST cyclones. The method is capable of relating this difference to diverse aspects of cyclone development, such as the tilt of the trough axis and altered static stability from LH. Further insights into the ETC development would be achieved if friction and cloud radiative effects can be isolated from $F_{RES}$. This is worthy since the NT cyclones develop and travel over the continent and large condensation occurs with the ST cyclones.

The ETC activities over East Asia exhibit a long-term trend as well as interannual variation (Wang et al., 2007; Chen et al., 2019; Lee et al., 2019). This study gives aid in understanding these temporal variations by quantitatively proposing the key positive and negative factors of the ETC development in the region. However, the present study is confined to the cold season ETCs. While there are only a few NT cyclones in summer, the ST cyclones are frequent in summer and are often responsible for heavy precipitation events (Park et al., 2020). Since diabatic contributions to the ETC development are well evaluated through the PV tendency inversion, extending the analysis to summertime ETCs could be an intriguing future study.

**Code Availability**

The analyses tools, including the PV tendency inversion code, are available from the authors upon request.

**Data Availability**

The ERA-Interim data are available online at https://www.ecmwf.int/en/forecasts/datasets/reanalysis-datasets era-interim, and the authors downloaded them from the ECMWF Data Server, using the ECMWF WebAPI.

**Author Contribution**

JMK and SWS designed the analyses. JMK developed the codes for the analyses and wrote the manuscript. SWS contributed to the interpretation of the results and the completion of the manuscript.



**Competing Interests**

The authors declare that they have no conflict of interest.

**Acknowledgement**

This work was supported by the National Research Foundation of Korea (NRF) grant funded by the Korean government (MSIT) (NRF2018R1A5A1024958).

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

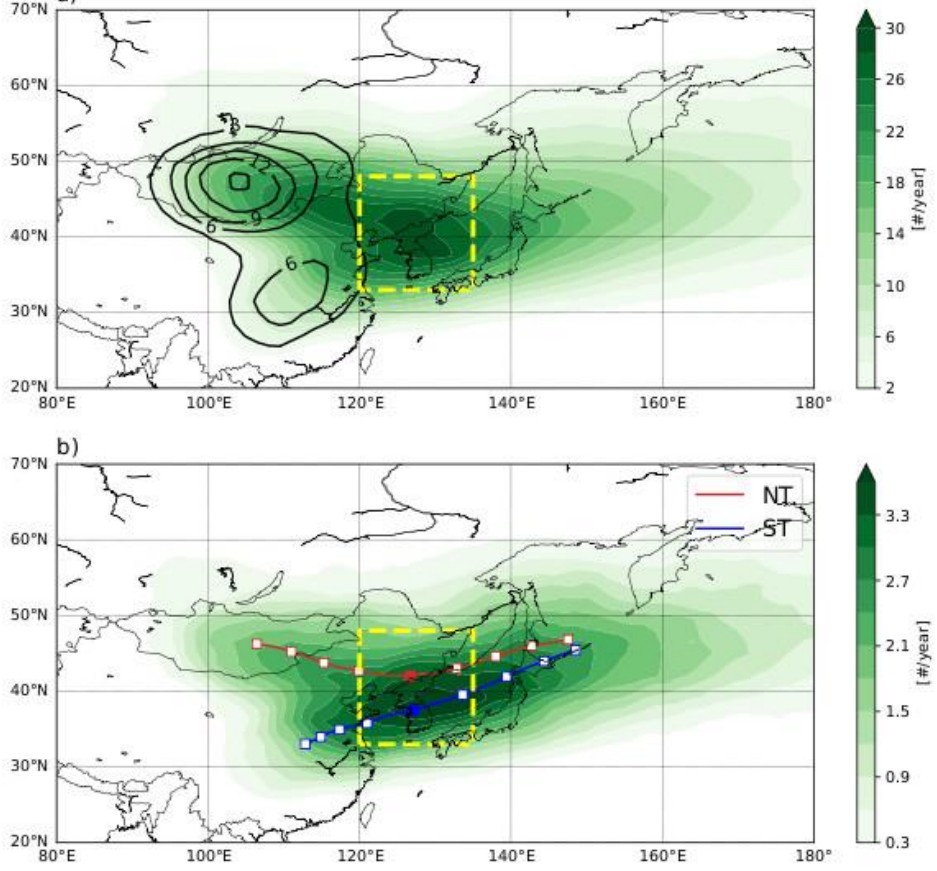



**Figure 1. (a) Track (shading, units: number per year) and cyclogenesis density (black, units: number per year) of the cold season ETCs passing the target domain (yellow box). (b) Track density (shading, units: number per year) of ETCs with top 10% intensification rate with average tracks of NT (red) and ST (blue) cyclones. The squares denote the average cyclone positions in 12-hour intervals from $t_{max}$ (filled square). Track and cyclogenesis density refer to**

**number of ETC tracks and genesis events within 555 km radius of each grid point.**

**Figure 2. (a) Cold season climatology of PV (shading, units: PVU) and wind (vectors, units: m s$^{-1}$) at 250 hPa. (b) Same as (a), but for integrated water vapor (shading, units: kg m$^{-2}$) and wind at 850 hPa (vectors, units: m s$^{-1}$).**





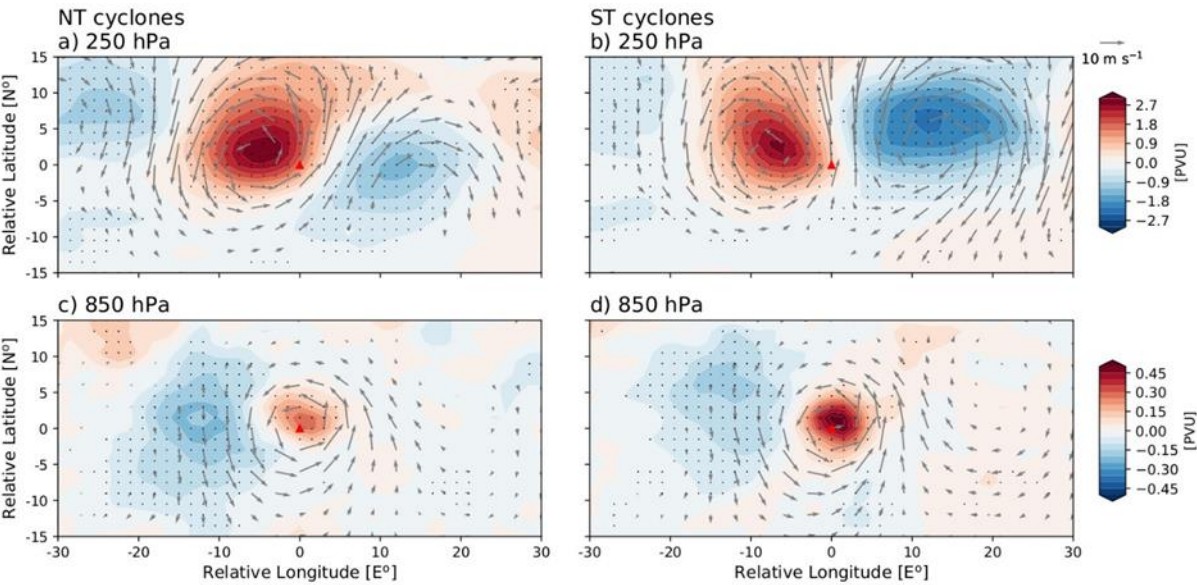

**Figure 3. (a) PV (shading, units: PVU) and wind (vectors, units: m s$^{-1}$) anomalies at 250 hPa with respect to the center of NT cyclones at t$_{max}$ (red triangle). (b) Same as (a), but for ST cyclones. (c, d) Same as (a, b), but at 850 hPa. The PV anomalies that are statistically significant at the 95% confidence level, based on the two-tailed Student's $t$-test, are dotted. Only statistically significant wind anomalies at the same confidence level are shown.**





**Figure 4. Vertical cross-section of the (a) $\frac{\partial q}{\partial t}$, (c) $-\mathbf{v} \cdot \nabla q$, (e) $-\omega \frac{\partial q}{\partial p}$, and (g) $Q_{LH}$ (shading, units: PVU (12h)$^{-1}$) with respect to the center of NT cyclones at $t_{max}$ (red triangle). (b, d, f, h) Same as (a, c, e, g), but for ST cyclones. Absolute values larger than 1 PVU (12h)$^{-1}$ are shown in additional contours in an 1-PVU (12h)$^{-1}$ interval. The values that are statistically significant at the 95% confidence level, based on the bootstrap random resampling test, are dotted (near-zero values are not dotted for conciseness).**

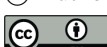



**Figure 5.** $\zeta_t$ from the inversion of (a) $\frac{\partial q}{\partial t}$, (c) $-\mathbf{v} \cdot \nabla q$, (e) $-\omega \frac{\partial q}{\partial p}$, (g) $Q_{LH}$, and (i) $-\frac{R_d}{p} \frac{\partial T}{\partial t}$ (shading, units: CVU $(12h)^{-1}$) with respect to the center of NT cyclones at $t_{max}$ (red triangle). (b, d, f, h) Same as (a, c, e, g), but for ST cyclones. The values that are statistically significant at the 95% confidence level, based on the bootstrap random resampling test, are dotted (near-zero values are not dotted for conciseness).

430

**Figure 6.** (a) Area-averaged $\zeta_t$ (bars, units: CVU $(12h)^{-1}$) from the inversions for NT cyclones at $t_{max}$ with 95% confidence intervals calculated from the bootstrap resampling method. The area-averaged $\zeta_t$ from reanalysis (pink





lines), sum of piecewise inversions (right black), and $F_{RES}$ (gray) are also shown. The values denoted to each bars

435 refer to the relative contribution (%) to the $\zeta_t$ from $\frac{\partial q}{\partial t}$. (b) Same as (a), but for ST cyclones.

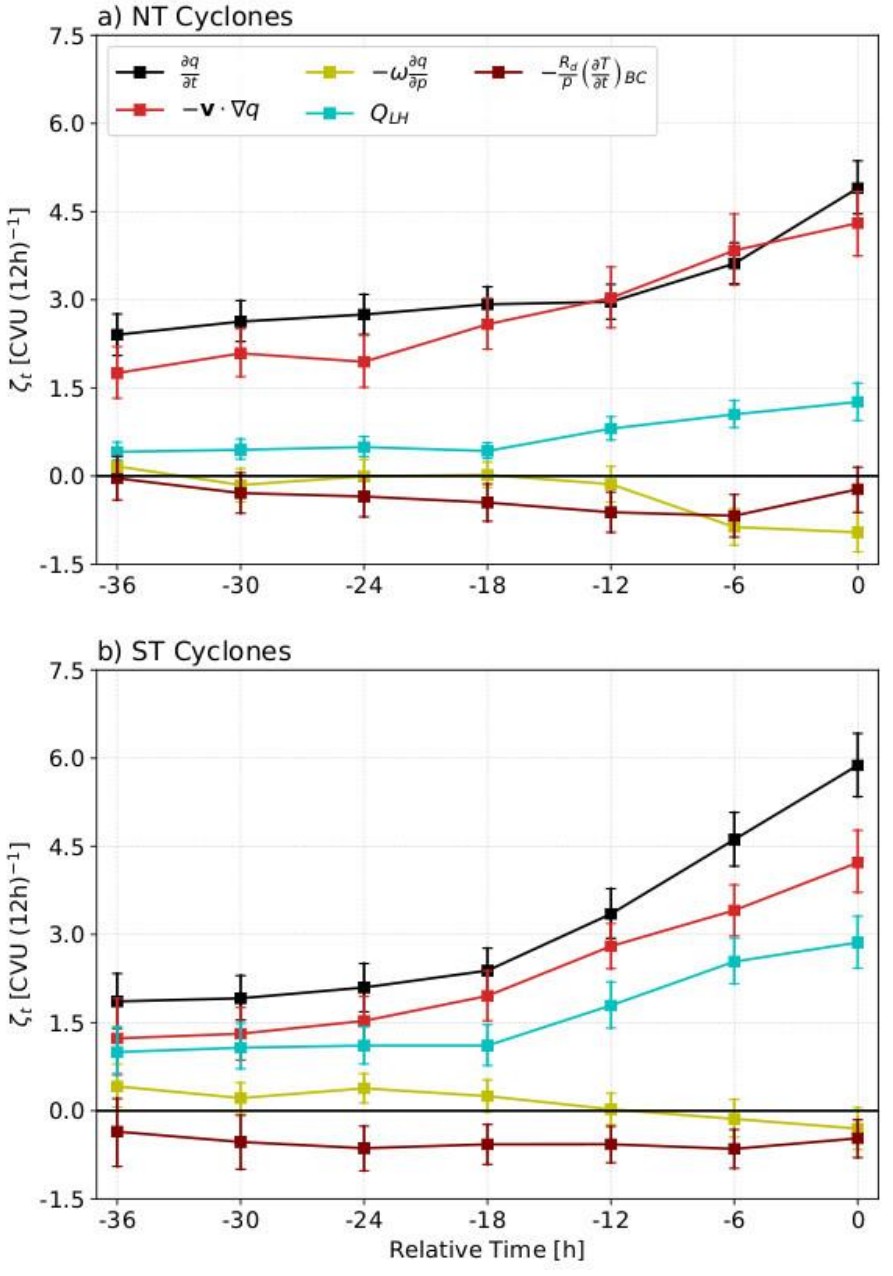

**Figure 7**. (a) Same as Fig. 6a, but from $t_{max}-36$ to $t_{max}$. (b) Same as (a), but for ST cyclones.



**Figure 8. (a) Same as Fig. 6a, but for decomposed advection terms. (b) Same as (a), but for ST cyclones. In (a) and (b), the left and right bars in the pair of bars represent upper- and lower-tropospheric contributions, respectively.**