# Peer review of "Development Processes of the East Asian Cyclones over the Korean Peninsula"

_Weather and Climate Dynamics, 2020_

## Referee Comment (RC1)

**The 1ˢᵗ Review Opinion for "Development Processes of the East Asian Cyclones over the Korean Peninsula" (WCD-2020-65)**

by Joonsuk M. Kang, Seok-Woo Son

(08 February 2021)

**General Comments:**

By using the six-hourly ERA-Interim data interpolated onto $1.5^o \times 1.5^o$ latitude-longitude grids, this manuscript aims to quantitatively evaluate the development processes of the extratropical cyclones (ETC) passing the Korean Peninsula from 1979 to 2017 by using the potential vorticity (PV) tendency equation. A feature tracking algorithm using the relative vorticity at 850 hPa is applied to objectively identify the so-called northern- and southern-track (NT and ST) cyclones affecting the Korean Peninsula region in the cold season (October to May). The dynamic and thermodynamic contributions to the development of these two categories of cyclones are then comparatively assessed from the PV tendency equation. The results suggested that East Asian cyclones passing the Korean Peninsula had different development processes depending on their tracks. Generally, the structure of this manuscript is fine, and its English language is better being non-native authors, as most sentences can be understood easily without any difficulties. However, it is necessary to indicate that, this manuscript at the present stage, didn't supply new and sharp insights into to deeply understand extratropical cyclones, and it is lack of sufficient explanation of physics for those two types of extratropical cyclones passing over the Korean Peninsula. Thus, it is very hard for referee to recommend this manuscript to be acceptable before the authors make clearer clarifications to the following key questions. It strongly suggests that this manuscript needs MAJOR REVISION before its potential publication in WCD.

**Major Comments:**

1. The background and methods of tracking ETCs were not introduced

adequately. There were a great number of methods of tracking ETCs. As indicated by Neu et al. (2013) "Identifying and tracking extratropical cyclones might seem, superficially, to be a straightforward activity, but in reality it is very challenging" (line 1-3 in right half, P529). The use of vorticity at 850 hPa for cyclone tracking is only one of 22 methods to identify ETCs as indicated by Neu et al. (2013) (see their Table 1 in P532). Different tracking methods may produce great quantitative differences in the total numbers of ETC (line 24-line 32 in right half page, P535) from 5~6 thousands to 21~28 thousands in two hemispheres. The present manuscript is lack of introducing the background as well as the advantage/disadvantage of using relative vorticity at 850 hPa for cyclone tracking.

2. The present title of this manuscript "Development Processes of the East Asian Cyclones over the Korean Peninsula" is not consistent with its content. More previous studies had indicated, and even the present authors admitted, that the development processes of ETCs involved in more complicated physics processes such as baroclinic process, upper-level trough, latent heating and so on. This manuscript only examines the development (intensifying/deepening) processes from one of various angles: PV perspective. Thus, it is suggested to use "PV Perspective of Development Processes of the East Asian Cyclones over the Korean Peninsula" or "Development Processes of the East Asian Cyclones over the Korean Peninsula: A Potential Vorticity Perspective" or other suitable title if the authors insisted on using PV analysis. It is much better to leave more rooms for other researchers.

3. The data and methods used in the present study is strongly argued. The present authors employed the six-hourly ERA-Interim data during the period from 1979 to 2017 which were interpolated onto $1.5^o \times 1.5^o$ (same data with K20), and the ETCs under investigation were identified on the 850-hPa relative vorticity field "Note that these ETCs fall into the categories of rapidly intensifying cyclones in K20" (line 89). Why did not the present authors use

more high-resolution data, for instance, ERA5 data (issued by ECMWF) with horizontal resolution $0.25^o \times 0.25^o$ and 1 hour interval? Or even FNL data (issued by NCEP) with horizontal resolution $1^o \times 1^o$ and 6 hour interval? As indicated by authors "The PV is calculated from horizontal winds and temperature by approximating partial differentials with second-order finite differencing" (line 65). What an error will be produced in the calculation of relative vorticity field at 850 hPa, then PV, from horizontal winds by approximating partial differential? As the target domain is $120^o$E-$135^o$E, $33^o$E-$48^o$N (line 78) with a region $15^o \times 15^o$, it suggests that whole domain only covers $5 \times 5$ PV values if the authors employed the "central finite differencing" scheme to calculate the PV from horizontal winds. How to calculate the geopotential tendency limited by the boundary conditions?

4. It is not convincing that two important references supporting WCD-2020-65 properly. The authors indicated at "In East Asia, cyclogenesis is remarked over Mongolia and East China (Chen et al., 1991; Adachi and Kimura, 2007 " (line 29). This citation plays quite important role for WCD-2020-65. We examined carefully the detailed information from words and figures in the aforementioned two papers (named as "Chen1991" and "AK2007", respectively), and found a great discrepancy between Fig. 2 of "Chen1991" (see Figure A) and Fig.1a of WCD-2020-65 (see Figure B). It is OK that "N region" is Figure A can be seen in Figure B correspondingly. But it is very strange that "S region" (the high cyclogenesis density region ) in Figure A can not be found in Figure B. This may be perhaps explained that they used different period data with different resolution. "Chen1991" used the twelve-hourly historic weather maps from 1958 to 1987, and $2.5^o \times 2.5^o$ latitude-longitude (coarse-resolution data) for cyclogenesis frequency counting. WCD-2020-65 used six-hourly $1.5^o \times 1.5^o$ ERA-Interim data (fine-resolution data) from 1979 to 2017. The overlapping period of these two data is INDEED 9 years (from 1979 to 1987). It is very hard

for referee to understand the important feature of "high cyclogenesis density region" in "Chen1991" (coarse-resolution data) disappeared in WCD-2020-65 (fine-resolution data). Is it suggested this great discrepancy hint that the method of using vorticity at 850 hPa for cyclone tracking is questionable? Moreover, the present referee failed to find "cyclogenesis is remarked over East China". In "Chen1991" paper, we didn't find the term "East China" but "East China Sea". In AK2007 paper, they mentioned "East China Sea" several times, but not "East China". It is strongly expected that the authors could clarify this issue.

[Figure]

**Figure A:** This figure was cited from Figure 2 in page 1409 of "Chen1991". "FIG.2 Annual number of cyclogenetic events ($10^{-2}$) per 2.5° quadrangle per month for the period 1958-87".

[Figure]

**Figure B:** This figure was cited from Figure 1a in page 15 of WCD-2020-65. "Figure 1. (a) Track (shading, units: number per year) and cyclogenesis density (black, units: number per year) of the cold season ETCs passing the target domain (yellow box)".

5. In the present study, the PV tendency equation is a very important tool for analysis. The authors also described this equation from term to term (line 95-103) as follows:

"The first term on the rhs of Eq. (1), representing the horizontal PV advection, describes the effects of the propagation and interaction of the upper-level trough and lower-level cyclonic circulation. The second term, representing the vertical PV advection, is physically related to the vertical change of adiabatic cooling which generally weakens ETC development. The third and fourth terms respectively stand for local PV changes from latent heating and other non conservative processes such as friction and cloud radiation". It is very strange that "in terms of the relative vorticity tendency resulting from the PV tendency inversion" (line 10). How about "relative vorticity tendency resulting from PV tendency inversion" ? If there exists a "relative vorticity tendency" term in PV tendency inversion, please describe it to all readers.

6. It is very hard for referee to understand the theory described from line 104 to line 130. What is(are) the scientific purpose(s) to establish "large theoretical framework" ? If the authors need the information of relative vorticity tendency $(\zeta_t)$ (line 182), it is easy to calculate this term according to the following relative vorticity equation:

$$\frac{\partial \zeta}{\partial t} = -\vec{V} \cdot \nabla(\zeta + f) - \omega \frac{\partial \zeta}{\partial p} - (\zeta + f)\nabla \cdot \vec{V} + \vec{k} \cdot (\frac{\partial \vec{V}}{\partial p} \times \nabla \omega) \qquad (4.21)$$

Please see details about the relative vorticity equation (4.21) in page 103 of Holton's book "An Introduction to Dynamic Meteorology" (4th edition).

7. Is it suitable to define the "cold season" from October to May (8 months of one year)? Usually, in mid-latitude region from 30°N to 50°N, May belongs to the early summer season. Are there any references from other scholars to support this definition about "cold season"?

**Specific Comments:**

1. Line 11-12, "It is quantified through inversion that the NT cyclones develop

87.9% dynamically and 6.2% thermodynamically. In contrast, the respective contributions to the ST cyclones are 71.8% and 43.5% for the ST cyclones". Sorry, we didn't understand the meaning of "percentage" in this sentence. Does it refer to the "number of ETS" or "the percentage of dynamic mechanism and thermodynamic mechanism"? Now the sum of them is not 100%.

2. Line 64, " ... specific humidity, and pressure velocity data during 39 years ...". It seems more appropriate to express "pressure velocity" as "vertical velocity in pressure coordinate".

3. Line 69, " ... the Hodges (1995,1999) algorithm ..." Here, a space is required after the comma in the sentence.

4. Line 70, "The spatial filtering is made to focus on synoptic-scale circulation". What kind of spatial filtering method is used in this paper?

5. Line 84, "the top 10% maximum intensification rate". Why did the authors select "top 10% maximum"?

6. Line 85, "The intensification rate is calculated as the 12-hour difference of relative vorticity". On which layer did the author calculate the relative vorticity?

7. Line 87-88 "The average tracks of these NT and ST cyclones are depicted in Fig.1b". How to define the average tracks of these NT and ST cyclones? What is(are) the scientific purpose(s) to do them? Unfortunately, we only find "one red track" and "one blue track" in Fig.1b (not many tracks).

8. Line 89, "Note that these ETCs fall into the categories of rapidly intensifying cyclones in K20.." There are two periods after the sentence.

9. Line 92, "the PV tendency inversion method (KS20)". We did not find the "KS20" paper at all. Even the "*Kang, J. M., and Son, S.-W.: Development processes of explosive cyclones over the Northwest Pacific: Potential vorticity perspective, J. Atmos. Sci., revised*" can be found, it is not suitable to work as a formal reference, because it is under revision.

10. Line 106, "The overbar denotes the mean, which is the monthly climatology

during the analysis period". The average physical quantity used by the authors in the calculation process is monthly mean, while the life cycle of extratropical cyclones is only about one week. Are these two different time scales suitable for study together ?

**Key References:**

Adachi, S., and F. Kimura, 2007: A 36-year climatology of surface cyclogenesis in East Asia using high-resolution reanalysis data, *SOLA*, **3**, 113-116.

Chen, S.-J., Y.-H. Kuo, P.-Z. Zhang, and Q.-F. Bai, 1991: Synoptic climatology of cyclogenesis over East Asia, 1958-1987. *Mon. Wea. Rev.*, **119**, 1407-1418.

Holton, J. R., 2004: An Introduction to Dynamic Meteorology (4th Edition), *Elsevier Academic Press*, 529.pp

Neu, U, and Coauthors, 2013: IMILAST: A Community Effort to Intercompare Extratropical Cyclone Detection and Tracking Algorithms, *Bull. Amer. Meteor. Soc.*, **94(4)**, 529-547.

https://journals.ametsoc.org/view/journals/bams/94/4/bams-d-11-00154.1.xml

---

## Referee Comment (RC2)

**Summary:**

The authors investigate the development processes of the cold-season East Asian Cyclones over the Korean Peninsula using a potential vorticity tendency analysis of cyclone-tracking composites. Through the detailed PV budget analysis, they reveal the different roles of the horizontal PV advection, vertical PV advection, latent heating release in the development of two groups of extratropical cyclones passing the Korean Peninsula (northern- and southern-track cyclones). They found that northern-track cyclones are dynamical dominant while the southern-track cyclones are both dynamical and thermodynamical driven.

**Recommendation:**

Several prior studies on the extratropical cyclones from potential vorticity tendency analysis have been predominantly statistical in nature. These studies put their focuses on oceanic extratropical cyclones either in North Pacific or North Atlantic. I believe that this study brings an important contribution by assessing the dynamical and thermodynamical processes in the development of continental extratropical cyclones. I thus recommend the authors to perform a major revision by considering the comments listed below.

**Major comments:**

The advantage of using the PV framework is that it provides a simple way to include the role of diabatic heating due to latent heating release and radiation. In this manuscript, the radiation and friction are put together into one term $F_{res}$ as in Eq. (1). And in the following sections, the contributions from radiation and frictions are not shown as well. However, as suggested in Tamarin and Kaspi 2016, the radiation contribution seems larger than the vertical PV advection in the cyclone development. Could the authors add some discussions on the estimation of radiation effects on the East Asian extratropical cyclones?

In Fig.8, the authors quantify the relative contributions of each component to the 850-hPa relative vorticity tendency from upper troposphere and lower troposphere. However, the detailed method for the algorithm and vertical decomposition is not described in the manuscript. Is it a piecewise PV inversion method in which the wind is decomposed from upper level and lower level? Could the authors explain the reason to choose the 600-hPa level to understand the behavior of 850-hPa relative vorticity? Please also specify the range of the upper-troposphere and lower-troposphere in line 257. For example, 175-600 hPa for upper troposphere and 600-875 hPa for lower troposphere. Is it a vertical average?

**Minor/Specific comments:**

Line 12: "… the respective contributions to the ST cyclones are 71.8% and 43.5% for the ST cyclones…" Two times of ST cyclones are found in this sentence. Maybe delete one of them?

Line 76: travels→travel

Line 79: is selected→are selected

Line 80: More than 25 ETCs impact the region in each along the two distinct ETC tracks. Could the author specify the time period (e.g. per year) to help the reader?

Line 195: …at a single level as in Figs. 4c and d…, perhaps the authors mean Figs. 5c and d?

Line 261: is derived→ are derived

Line 291: exist→ exists, then than→ then

---

## Author Comment (AC1)

Response to Reviewer 1 for

**Development Processes of the East Asian Cyclones over the Korean**

**Peninsula: A Potential Vorticity Perspective**

submitted to *Weather and Climate Dynamics*

The authors thank the reviewer for carefully reviewing the manuscript. The reviewer's comments are answered in detail, after the general comments from the reviewer. This study is motivated and established upon three key references. Lee et al. (2019) presents an updated climatology of East Asian ETCs by applying a feature tracking algorithm to 850-hPa relative vorticity field. Kang et al. (2020) reports the mechanisms responsible for the (feature-tracked) ETC developments around the Korean Peninsula through composite analysis. The quantitative method (PV tendency inversion), which is used to evaluate the processes contributing to ETC development, is introduced in Kang and Son (2021).

**General Comments**

*By using the six-hourly ERA-Interim data interpolated onto 1.5 o × 1.5 o latitude-longitude grids, this manuscript aims to quantitatively evaluate the development processes of the extratropical cyclones (ETC) passing the Korean Peninsula from 1979 to 2017 by using the potential vorticity (PV) tendency equation. A feature tracking algorithm using the relative vorticity at 850 hPa is applied to objectively identify the so-called northern- and southern-track (NT and ST) cyclones affecting the Korean Peninsula region in the cold season (October to*

*May). The dynamic and thermodynamic contributions to the development of these two categories*

*of cyclones are then comparatively assessed from the PV tendency equation. The results*

*suggested that East Asian cyclones passing the Korean Peninsula had different development*

*processes depending on their tracks. Generally, the structure of this manuscript is fine, and its*

*English language is better being non-native authors, as most sentences can be understood easily*

*without any difficulties. However, it is necessary to indicate that, this manuscript at the present*

*stage, didn't supply new and sharp insights into to deeply understand extratropical cyclones, and*

*it is lack of sufficient explanation of physics for those two types of extratropical cyclones passing*

*over the Korean Peninsula. Thus, it is very hard for referee to recommend this manuscript to be*

*acceptable before the authors make clearer clarifications to the following key questions. It*

*strongly suggests that this manuscript needs MAJOR REVISION before its potential publication*

*in WCD.*
* * *
**Major Comments**

**(1)**

**Reviewer:** *The background and methods of tracking ETCs were not introduced adequately.*

*There were a great number of methods of tracking ETCs. As indicated by Neu et al. (2013)*

*"Identifying and tracking extratropical cyclones might seem, superficially, to be a*

*straightforward activity, but in reality it is very challenging" (line 1-3 in right half, P529). The*

*use of vorticity at 850 hPa for cyclone tracking is only one of 22 methods to identify ETCs as*

*indicated by Neu et al. (2013) (see their Table 1 in P532). Different tracking methods may*

*produce great quantitative differences in the total numbers of ETC (line 24-line 32 in right half*

*page, P535) from 5~6 thousands to 21~28 thousands in two hemispheres. The present*

*manuscript is lack of introducing the background as well as the advantage/disadvantage of using*

*relative vorticity at 850 hPa for cyclone tracking.*

**Response:** The selection of the tracking variable and pressure level follows Lee et al. (2019) and

Kang et al. (2020). The advantages and disadvantages of using relative vorticity at 850 hPa can

be found in the Sections 1 and 2 of Lee et al. (2019). In the revised manuscript, we note that we

are using one of the various tracking algorithms, and that more details can be found in Lee et al.

(2019).

**Lines 69-71:** Among diverse methods for objective ETC tracking (Neu et al., 2013), the Hodges

(1995, 1999) algorithm is employed in this study. This algorithm is applied to the 850-hPa

relative vorticity field, subject to spatial filtering with total wavenumber 5 to 42, at six-hourly

intervals as in previous studies (Cho et al., 2018; Lee et al., 2019; K20).

**Lines 78-79:** See Lee et al. (2019) for further details, including the advantages and

disadvantages of this method in East Asian domain.

**(2)**

**Reviewer:** *The present title of this manuscript "Development Processes of the East Asian*

*Cyclones over the Korean Peninsula" is not consistent with its content. More previous studies*

*had indicated, and even the present authors admitted, that the development processes of ETCs*

*involved in more complicated physics processes such as baroclinic process, upper-level trough,*

*latent heating and so on. This manuscript only examines the development (intensifying/deepening) processes from one of various angles: PV perspective. Thus, it is suggested to use "PV Perspective of Development Processes of the East Asian Cyclones over the Korean Peninsula" or "Development Processes of the East Asian Cyclones over the Korean Peninsula: A Potential Vorticity Perspective" or other suitable title if the authors insisted on using PV analysis. It is much better to leave more rooms for other researchers.*

**Response:** Thank you for the suggestions. We changed the title to:

"Development Processes of the East Asian Cyclones over the Korean Peninsula: A Potential Vorticity Perspective".

**(3)**

**Reviewer:** *The data and methods used in the present study is strongly argued. The present authors employed the six-hourly ERA-Interim data during the period from 1979 to 2017 which were interpolated onto 1.5 o×1.5 o (same data with K20), and the ETCs under investigation were identified on the 850-hPa relative vorticity field "Note that these ETCs fall into the categories of rapidly intensifying cyclones in K20" (line 89). Why did not the present authors use more high-resolution data, for instance, ERA5 data (issued by ECMWF) with horizontal resolution 0.25 o × 0.25 o and 1 hour interval? Or even FNL data (issued by NCEP) with horizontal resolution 1 o×1 o and 6 hour interval? As indicated by authors "The PV is calculated from horizontal winds and temperature by approximating partial differentials with second-order finite differencing" (line 65). What an error will be produced in the calculation of relative vorticity field at 850 hPa, then PV, from horizontal winds by approximating partial*

*differential? As the target domain is 120 oE-135 oE, 33 oE- 48 oN (line 78) with a region 15 o×*

*15 o , it suggests that whole domain only covers 5 × 5 PV values if the authors employed the*

*"central finite differencing" scheme to calculate the PV from horizontal winds. How to calculate*

*the geopotential tendency limited by the boundary conditions?*

**Response:** The PV on an isobaric surface is expressed as follows.

$$q = -g \frac{\partial \theta}{\partial p} \left( \frac{\partial v}{\partial x} - \frac{\partial u}{\partial y} + f \right) + g \left( \frac{\partial \theta}{\partial x} \frac{\partial v}{\partial p} - \frac{\partial \theta}{\partial y} \frac{\partial u}{\partial p} \right) \tag{R1}$$

In the present study, Eq. (R1) is computed by second-order finite differencing with $1.5^\circ \times 1.5^\circ$

data.

It turns out that synoptic scale PV is not very sensitive to the data resolution in ERA-

Interim. Figure R1 illustrates the vertical cross-section composite of PV at $t_{max}$ for NT and ST

cyclones. The first row (Figs. R1a and b) shows the PV archived directly from the ERA-Interim

database, whereas the second and third rows (Figs. R1c–f) show the PV computed from Eq. (R1)

with two different horizontal resolutions (i.e., $1.5^\circ$ and $0.75^\circ$). In all three rows, the westwardtilted PV structure from the surface to tropopause is evident. More importantly, the PV values

are quantitatively similar. This confirms that the second-order finite differencing does not lead to

misinterpretation of PV features related to the ETCs. The numerical errors arising from secondorder finite differencing are not influential in our synoptic-scale analysis. Note that the abnormal

PV values at the bottom left corners of Figs. R1a and b, resulting from extrapolation errors under

the Tibetan Plateau, are effectively removed by computing PV from extrapolated winds and

temperatures (Figs. R1c–f).

ERA-Interim is used instead of ERA5 is used in this study to be consistent with Lee et al.

(2019) and Kang et al. (2020). As stated earlier, this work is an extension of Lee et al. (2019) and

Kang et al. (2020), providing more quantitative aspects of East Asian ETC development. We admit that the quantitative result could be slightly different if high-resolution ERA5 data is utilized. However, the overall results, i.e., the relative importance of dynamic and thermodynamic processes, would not change, as inferred from the negligible difference between 1.5$^{\circ}$ and 0.75$^{\circ}$ resolutions in Fig. R1 (compare second and third rows).

The target domain in Fig. 1 is used for sampling ETCs that pass through it. The inversion is carried out in a wider domain, spanning 60$^{\circ}$ zonally and 30$^{\circ}$ meridionally about the ETC center (horizontal margins of Fig. 5). At each level, the domain has 39×19 interior PV tendency values, and 41×21 geopotential tendency is returned after the inversion (see appendix B of KS21).

[Figure]

**Figure R1. (a–c)** Vertical cross-section of PV (shading, units: PVU) at $t_{max}$ for NT cyclones. The cross-sections are made from **(a)** archived directly from ERA-Interim database (1.5$^o$×1.5$^o$), **(b, c)** calculated by Eq. (R1) with **(b)** 1.5$^o$×1.5$^o$ data, and **(c)** 0.75$^o$×0.75$^o$ data **(d–f)** Same as **(a–c)**, but for ST cyclones.

**(4)**

**Reviewer**: *It is not convincing that two important references supporting WCD-2020-65 properly. The authors indicated at "In East Asia, cyclogenesis is remarked over Mongolia and East China (Chen et al., 1991; Adachi and Kimura, 2007"(line 29). This citation plays quite important role for WCD-2020-65. We examined carefully the detailed information from words and figures in the aforementioned two papers (named as "Chen1991" and "AK2007", respectively), and found a great discrepancy between Fig. 2 of "Chen1991" (see Figure A) and Fig.1a of WCD-2020-65 (see Figure B). It is OK that "N region" is Figure A can be seen in Figure B correspondingly. But it is very strange that "S region" (the high cyclogenesis density region ) in Figure A can not be found in Figure B. This may be perhaps explained that they used different period data with different resolution. "Chen1991" used the twelve-hourly historic weather maps from 1958 to 1987, and 2.5 o × 2.5 o latitude-longitude (coarse-resolution data) for cyclogenesis frequency counting. WCD-2020-65 used six-hourly 1.5 o × 1.5 o ERA-Interim data (fine-resolution data) from 1979 to 2017. The overlapping period of these two data is INDEED 9 years (from 1979 to 1987). It is very hard for referee to understand the important feature of "high cyclogenesis density region" in "Chen1991" (coarse-resolution data) disappeared in WCD-2020-65 (fine-resolution data). Is it suggested this great discrepancy hint that the method of using vorticity at 850 hPa for cyclone tracking is questionable? Moreover, the present referee failed to find "cyclogenesis is remarked over East China". In "Chen1991" paper, we didn't find the term "East China" but "East China Sea". In AK2007 paper, they mentioned "East China Sea" several times, but not "East China". It is strongly expected that the authors could clarify this issue.*

**Response:** While AK2007 utilizes sea-level pressure (SLP) for objective ETC tracking, Chen1991 relies on manual tracking on surface weather maps. In other words, both of them are based on SLP field. Since the relative vorticity at 850-hPa pressure level is used in this study, the resulting ETC properties could be different from those reported in AK2007 and Chen1991.

The difference between AK2007/Chen1991 and the present study mostly results from the difference in reference variables. As described in Hoskins and Hodges (2002) and Lee et al. (2019), the relative vorticity field, compared to SLP field, captures a relatively smaller-scale perturbation and allows and early detection of the surface cyclone. In fact, a strong cyclonic vorticity (momentum field) is often followed by a minimum SLP (mass field) with a time lag. Their relationship is well documented in Hoskins and Hodges (2002) who compared the cyclogenesis statistics achieved from SLP and relative vorticity tracking. Figure R2 shows their Figs. 5c and 6c. In North China/Mongolia, a cyclogenesis peak, derived from SLP tracking (left panel), appears downstream of that from relative vorticity tracking (right panel). Likewise, a cyclogenesis peak in East China in the left panel appears further downstream of that in the right panel, near the coasts of the East China Sea. This difference explains why the SLP-based cyclogenesis region in AK2007 and Chen1991 refers 'East China Sea', whereas that in Lee et al. (2019) refers 'East China'.

Besides, the discrepancy between Fig. 2 of Chen1991 and our result is due to the difference in sampled ETCs. The track density and cyclogenesis shown in Fig. 1a of the manuscript are calculated for the ETCs that pass the Korean peninsula, not for all ETCs in East Asia. The cyclones generated at "S region", which is located southeast of the Korean Peninsula, are unlikely to pass through the target domain. We clarified this point in the revised manuscript as follows.

**Lines 80-83:** From the tracking algorithm, the ETCs passing the Korean Peninsula (120–135ºE, 33–48ºN; yellow box in Fig. 1) are selected. As in K20, only the ETCs that are generated west of 120ºE are considered to focus on their development processes while traveling eastward through the target region. The track density of the selected ETCs in the cool season (October–May) is illustrated in Fig. 1a.

[Figure]

**Figure R2. (left)** Fig. 5c (SLP) and **(right)** Fig. 6c (relative vorticity) in Hoskins and Hodges (2002).

**(5)**

**Reviewer:** *In the present study, the PV tendency equation is a very important tool for analysis. The authors also described this equation from term to term (line 95-103) as follows: "The first term on the rhs of Eq. (1), representing the horizontal PV advection, describes the effects of the propagation and interaction of the upper-level trough and lower-level cyclonic circulation. The second term, representing the vertical PV advection, is physically related to the vertical change of adiabatic cooling which generally weakens ETC development. The third and fourth terms*

*respectively stand for local PV changes from latent heating and other non conservative processes such as friction and cloud radiation". It is very strange that "in terms of the relative vorticity tendency resulting from the PV tendency inversion" (line 10). How about "relative vorticity tendency resulting from PV tendency inversion" ? If there exists a "relative vorticity tendency" term in PV tendency inversion, please describe it to all readers.*

**Response:** Thanks for pointing this out. It is clarified in the revised manuscript as below.

**Lines 10-11:** With respect to the 850-hPa geostrophic relative vorticity tendency resulting from the PV tendency inversion, …

**(6)**

**Reviewer:** *It is very hard for referee to understand the theory described from line 104 to line 130. What is(are) the scientific purpose(s) to establish "large theoretical framework" ? If the authors need the information of relative vorticity tendency ($\zeta t$ ) (line 182), it is easy to calculate this term according to the following relative vorticity equation:*

$$\frac{\partial \zeta}{\partial t} = -\vec{V} \cdot \nabla (\zeta + f) - \omega \frac{\partial \zeta}{\partial p} - (\zeta + f)\nabla \cdot \vec{V} + \vec{k} \cdot (\frac{\partial \vec{V}}{\partial p} \times \nabla \omega) \qquad (4.21)$$

*Please see details about the relative vorticity equation (4.21) in page 103 of Holton's book "An Introduction to Dynamic Meteorology" (4 th edition).*

**Response:** The purpose of establishing such a theoretical framework is elaborated in detail in KS21. In short, there are at least two reasons why the method proposed in this study should be used instead of (4.21) from Holton (2004).

(i) Thermodynamic contributions are not measured from (4.21). Therefore, potential vorticity equation should be accounted for, which can be derived from (4.21) and the thermodynamic energy equation.

(ii) The terms on the right-hand-side of (4.21) are calculated at a single pressure level. The influences of forcings from other levels are not explicitly assessable.

It is desirable to have a method that utilizes potential vorticity equation (Eq. (1) in the manuscript) and enables to gauge the effects from forcings at different levels on the cyclone development. The latter can be done with the inversion technique in this study.

**(7)**

**Reviewer:** *Is it suitable to define the "cold season" from October to May (8 months of one year)? Usually, in mid-latitude region from 30 oN to 50 oN, May belongs to the early summer season. Are there any references from other scholars to support this definition about "cold season"?*

**Response:** Roebber (1984) defines eight months of a year (September–April) as cold season, which has almost the same duration, but has different months included. However, considering the reviewer's concern, we chose to use the term 'cool season' instead of 'cold season' following Ren et al. (2010).

**Minor Comments**

**(1)**

**Reviewer:** *Line 11-12, "It is quantified through inversion that the NT cyclones develop 87.9% dynamically and 6.2% thermodynamically. In contrast, the respective contributions to the ST cyclones are 71.8% and 43.5% for the ST cyclones". Sorry, we didn't understand the meaning of "percentage" in this sentence. Does it refer to the "number of ETS" or "the percentage of dynamic mechanism and thermodynamic mechanism" ? Now the sum of them is not 100%.*

**Response:** We clarified these issues in the revised version of the manuscript.

**Lines 10-13**: With respect to the 850-hPa geostrophic relative vorticity tendency resulting from the PV tendency inversion, it is quantified that the NT cyclones develop 87.9% dynamically and 6.2% thermodynamically. In contrast, the respective contributions are 71.8% and 43.5% for the ST cyclones. The excessive or insufficient contributions are complemented by non-explicit processes.

**(2)**

**Reviewer:** *Line 64, " ... specific humidity, and pressure velocity data during 39 years ...". It seems more appropriate to express "pressure velocity" as "vertical velocity in pressure coordinate*

**Response:** Modified as the reviewer's suggestion in the revised manuscript.

**(3)**

**Reviewer:** *Line 69, "... the Hodges (1995,1999) algorithm ..." Here, a space is required after the comma in the sentence.*

**Response:** Corrected in the revised manuscript.

**(4)**

**Reviewer:** *Line 70, "The spatial filtering is made to focus on synoptic-scale circulation". What kind of spatial filtering method is used in this paper?"*

**Response:** The selection of specific wavenumbers (from 5 to 42) in the total wavenumber space refers to spatial filtering. We made this clearer in the revised manuscript.

**Lines 71-72:** This algorithm is applied to the 850-hPa relative vorticity field, subject to spatial filtering with total wavenumber 5 to 42, at six-hourly intervals as in previous studies (Cho et al., 2018; Lee et al., 2019; K20).

**(5)**

**Reviewer:** *Line 84, "the top 10% maximum intensification rate". Why did the authors select "top 10% maximum".?*

**Response:** We followed K20 who defined the rapidly intensifying cyclones using the top 10% maximum intensification rate. This threshold is somewhat arbitrary. However, the same results are obtained even when the top 20% is analyzed. Figure R3 shows the vertical cross-section of PV tendency of the top 20% NT and ST cyclones. Although the PV tendencies are generally weaker than those in Figs. 4a and b in the manuscript, the overall structures are similar. Most importantly, the PV tendency in the upper level is stronger for NT cyclones, whereas that in the lower level is larger for ST cyclones, consistent with the results shown in this study.

[Figure]

**Figure R3.** Same as Figs. 4a and b in the manuscript, but for ETCs with the top 20% maximum intensification rate.

**(6)**

**Reviewer:** *Line 85, "The intensification rate is calculated as the 12-hour difference of relative vorticity". On which layer did the author calculate the relative vorticity?*

**Response:** The relative vorticity at 850 hPa is used as in ETC tracking. We made this clearer in the revised manuscript.

**Lines 88-89:** Here, the intensification rate is calculated as the 12-hour difference of relative vorticity at 850 hPa, …

**(7)**

**Reviewer:** *Line 87-88 "The average tracks of these NT and ST cyclones are depicted in Fig.1b". How to define the average tracks of these NT and ST cyclones? What is(are) the scientific*

*purpose(s) to do them? Unfortunately, we only find "one red track" and "one blue track" in*

*Fig.1b (not many tracks).*

**Response:** As noted in the text and the figure caption, the red and blue tracks represent the average tracks of NT and ST cyclones. They are calculated by connecting average ETC positions in 6-hourly intervals from $t_{max}$. This is further clarified in the revised manuscript.

**Lines 90-92:** To visualize how these NT and ST cyclones propagate on average, their average tracks are calculated by connecting the average ETC positions in six-hourly intervals with respect to $t_{max}$. The average cyclone tracks are depicted in Fig. 1b, where NT and ST cyclones propagate northeastward after $t_{max}$.

**(8)**

**Reviewer:** *Line 89, "Note that these ETCs fall into the categories of rapidly intensifying cyclones in K20.." There are two periods after the sentence.*

**Response:** Corrected in the revised manuscript.

**(9)**

**Reviewer:** *Line 92, "the PV tendency inversion method (KS20)". We did not find the "KS20" paper at all. Even the "Kang, J. M., and Son, S.-W.: Development processes of explosive cyclones over the Northwest Pacific: Potential vorticity perspective, J. Atmos. Sci., revised" can be found, it is not suitable to work as a formal reference, because it is under revision.*

**Response:** KS20 (now KS21) is recently accepted for publication at the *Journal of the Atmospheric Sciences* and will be available online soon.

**(10)**

**Reviewer:** *Line 106, "The overbar denotes the mean, which is the monthly climatology7 during the analysis period". The average physical quantity used by the authors in the calculation process is monthly mean, while the life cycle of extratropical cyclones is only about one week. Are these two different time scales suitable for study together ?*

**Response:** The mean state is set as monthly climatology to suit the purpose of investigating the influence of the background flow, which varies during the cool season.

**References not in manuscript**

Ren, X., Yang, X., & Chu, C. (2010). Seasonal variations of the synoptic-scale transient eddy activity and polar front jet over East Asia. *Journal of climate*, *23*(12), 3222-3233.

Roebber, P. J. (1984). Statistical analysis and updated climatology of explosive cyclones. *Monthly Weather Review*, *112*(8), 1577-1589.

---

## Author Comment (AC2)

Response to Reviewer 2 for

**Development Processes of the East Asian Cyclones over the Korean Peninsula**

submitted to *Weather and Climate Dynamics*
* * *
The authors thank the reviewer for carefully reviewing the manuscript. The reviewer's

comments are answered in detail, after the general comments from the reviewer.

**General Comments**

*The authors investigate the development processes of the cold-season East Asian Cyclones over*

*the Korean Peninsula using a potential vorticity tendency analysis of cyclone-tracking*

*composites. Through the detailed PV budget analysis, they reveal the different roles of the*

*horizontal PV advection, vertical PV advection, latent heating release in the development of two*

*groups of extratropical cyclones passing the Korean Peninsula (northern- and southern-track*

*cyclones). They found that northern-track cyclones are dynamical dominant while the southern-*

*track cyclones are both dynamical and thermodynamical driven.*

*Several prior studies on the extratropical cyclones from potential vorticity tendency analysis*

*have been predominantly statistical in nature. These studies put their focuses on oceanic*

*extratropical cyclones either in North Pacific or North Atlantic. I believe that this study brings*

*an important contribution by assessing the dynamical and thermodynamical processes*

*in the development of continental extratropical cyclones. I thus recommend the authors to*

*perform a major revision by considering the comments listed below.*

**Major Comments**

**(1)**

**Reviewer:** *The advantage of using the PV framework is that it provides a simple way to include the role of diabatic heating due to latent heating release and radiation. In this manuscript, the radiation and friction are put together into one term Fres as in Eq. (1). And in the following sections, the contributions from radiation and frictions are not shown as well. However,as suggested in Tamarin and Kaspi 2016, the radiation contribution seems larger than the vertical PV advection in the cyclone development. Could the authors add some discussions on the estimation of radiation effects on the East Asian extratropical cyclones?*

**Response:** We thank the reviewer for the constructive suggestion. The PV tendency from radiative heating, $Q_{RAD}$, can be expressed as follows.

$$Q_{RAD} = -g(\zeta + f)\frac{\partial \dot{\theta}_{RAD}}{\partial p}$$

Here, $\dot{\theta}_{RAD}$ is the radiative heating obtained from the ERA-Interim model. Figures R1a and b show the vertical cross-sections of the radiative heating for NT and ST cyclones at $t_{max}$. The overall pattern is dominated by longwave cooling in the upper troposphere. This cooling can contribute to diabatic PV reduction in mid to lower troposphere.

The $Q_{RAD}$ at 850 hPa is shown in Figs. R2a and b. Unlike Tamarin and Kaspi (2016), it is considerably smaller than $-\omega\frac{\partial q}{\partial p}$ for both NT and ST cyclones (compare Figs. R2a,b and R2c,d).

This is also clear from the vertical cross-section shown in Fig. R3. Owing to this minor forcing, we decided not to include the analysis on radiative heating. This finding is briefly discussed in the revised manuscript as follows.

**Lines 316-320**: A preliminary analysis, based on the radiative heating of the ERA-Interim model data, reveals a weak but statistically significant radiative cooling (mostly due to longwave cooling) indeed appears in the upper troposphere above the level of maximum LH. However, its contributions to the PV tendency for both NT and ST cyclones are much smaller than $Q_{LH}$ and the advection terms (not shown). Further analyses using numerical models could be useful.

[Figure]

**Figure R1.** Vertical cross-section of total radiative heating (shading, units: K (12h)$^{-1}$) with respect to the center of **(a)** NT and **(b)** ST cyclones at t$_{max}$. Total radiative heating at 850 hPa (shading, units: K (12h)$^{-1}$) with respect to the center of **(c)** NT and **(d)** ST cyclones at t$_{max}$. In **(a)** and **(b)**, the vertical cross-section of LH is also shown in black contours (units: K (12h)$^{-1}$).

[Figure]

**Figure R2.** PV tendency from $Q_{RAD}$ (shading, units: PVU (12h)$^{-1}$) with respect to the center of **(a)** NT and **(b)** ST cyclones at t$_{max}$. **(c, d)** Same as **(a, b)**, but for $-\omega\frac{\partial q}{\partial p}$.

[Figure]

**Figure R3.** Vertical cross-section $Q_{RAD}$ (shading, units: PVU $(12h)^{-1}$) with respect to the center of **(a)** NT and **(b)** ST cyclones at $t_{max}$. **(c, d)** Same as **(a, b)**, but for $-\omega\frac{\partial q}{\partial p}$.

**(2)**

**Reviewer:** *In Fig.8, the authors quantify the relative contributions of each component to the 850-hPa relative vorticity tendency from upper troposphere and lower troposphere. However, the detailed method for the algorithm and vertical decomposition is not described in the manuscript. Is it a piecewise PV inversion method in which the wind is decomposed from upper level and lower level? Could the authors explain the reason to choose the 600-hPa level to understand the behavior of 850-hPa relative vorticity? Please also specify the range of the upper-troposphere and lower-troposphere in line 257. For example, 175-600 hPa for upper troposphere and 600-875 hPa for lower troposphere. Is it a vertical average?*

**Response:** Thank you for pointing this out. We have made it clearer in the revised manuscript as follows.

**Lines 262-265:** The decomposition of upper- and lower-tropospheric advection is done by setting the lower-level (875–600 hPa) PV tendency to zero when inverting upper-tropospheric (600–175 hPa) PV tendency, and vise versa. The 600-hPa reference level is chosen since positive PV tendency from horizontal advection in the upper troposphere intrudes down to this level (Figs. 4c and d), showing distinctive feature from the levels below.

**Minor Comments**

**(1)**

**Reviewer:** Line 12: "… the respective contributions to the ST cyclones are 71.8% and 43.5% for the ST cyclones…" Two times of ST cyclones are found in this sentence. Maybe delete one of them?

**Response:** Corrected in the revised manuscript.

**(2)**

**Reviewer:** Line 76: travels→travel

**Response:** Corrected in the revised manuscript.

**(3)**

**Reviewer:** Line 79: is selected→are selected

**Response:** Corrected in the revised manuscript.

**(4)**

**Reviewer:** Line 80: More than 25 ETCs impact the region in each along the two distinct ETC tracks. Could the author specify the time period (e.g. per year) to help the reader?

**Response:** Modified as reviewer's suggestion in the revised manuscript.

**(5)**

**Reviewer:** Line 195: …at a single level as in Figs. 4c and d…, perhaps the authors mean Figs. 5c and d?

**Response:** This sentence has been modified as follows.

**Lines 199-201:** This result clearly indicates that diagnosing the PV tendency at a particular level is insufficient at gauging its effect on ETC development, highlighting the advantage of inversion calculation.

**(6)**

**Reviewer:** Line 261: is derived→ are derived

**Response:** Corrected in the revised manuscript.

**(7)**

**Reviewer:** Line 291: exist→ exists, then than→ then

**Response:** In this sentence, 'at then' refers to 'at the initial stages'.